# A zwitterionic near-infrared fluorophore for real-time ureter identification during laparoscopic abdominopelvic surgery

Kim S. de Valk[1,2,4], Henricus J. Handgraaf [2,4], Marion M. Deken[2], Babs G. Sibinga Mulder[2], Adrianus R. Valentijn[2], Anton G. Terwisscha van Scheltinga[2], Joeri Kuil[2], Michiel J. van Esdonk[1], Jaap Vuijk[2], Rob F. Bevers[2], Koen C. Peeters[2], Fabian A. Holman[2], John V. Frangioni[3], Jacobus Burggraaf[1] & Alexander L. Vahrmeijer[2]

Iatrogenic injury of the ureters is a feared complication of abdominal surgery. Zwitterionic near-infrared fluorophores are molecules with geometrically-balanced, electrically-neutral surface charge, which leads to renal-exclusive clearance and ultralow non-specific background binding. Such molecules could solve the ureter mapping problem by providing real-time anatomic and functional imaging, even through intact peritoneum. Here we present the first-in-human experience of this chemical class, as well as the efficacy study in patients undergoing laparoscopic abdominopelvic surgery. The zwitterionic near-infrared fluorophore ZW800-1 is safe, has pharmacokinetic properties consistent with an ideal blood pool agent, and rapid elimination into urine after a single low-dose intravenous injection. Visualization of structure and function of the ureters starts within minutes after ZW800-1 injection and lasts several hours. Zwitterionic near-infrared fluorophores add value during laparoscopic abdominopelvic surgeries and could potentially decrease iatrogenic urethral injury. Moreover, ZW800-1 is engineered for one-step covalent conjugatability, creating possibilities for developing novel targeted ligands.

[1] Centre for Human Drug Research, Leiden, The Netherlands. [2] Leiden University Medical Center, Leiden, The Netherlands. [3] Curadel, LLC, Marlborough, MA, USA. [4] These authors contributed equally: Kim. S. de Valk, Henricus. J. Handgraaf. Correspondence and requests for materials should be addressed to K.S.d.V. (email: k.s.de_valk@lumc.nl)

ntraoperative near-infrared (NIR) fluorescence imaging has evolved rapidly over the past decade. The technology permits detection of specific targets, such as malignant cells, nerves, blood vessels, and lymph nodes, in real-time during surgery[1]. Because NIR wavelengths exhibit reduced scattering, absorption, and autofluorescence compared to visible wavelengths, NIR fluorophores permit detection of targets through millimeters of blood and tissue with high sensitivity.

However, a fundamental problem with NIR fluorescence imaging is that conventional NIR fluorophores are polysulfonated, and highly anionic, in order to shield the central hydrophobic resonance structure and improve solubility, and thus exhibit non-specific uptake in tissues and organs after intravenous (IV) injection. This results in high background fluorescence and consequently a lower signal-to-background ratio (SBR). Another problem is that amphiphiles, like indocyanine green (ICG), are rapidly (blood half-life ≈3 min) and exclusively cleared by the liver after IV injection, thus contaminating the bile and compromising imaging of the gastrointestinal tract. The same applies for other anionic fluorophores, such as IRDye800CW[2].

To solve these problems, a chemical class of geometrically balanced, electrically neutral, polyionic polymethine indocyanines (zwitterionic for short) NIR fluorophores were developed[3,4]. Having strong charge (sulfonates and quaternary amines) that is balanced electrically and geometrically over the surface of the molecule, zwitterionic NIR fluorophores are self-shielding and exhibit extremely low non-specific binding and tissue uptake in vivo after IV injection.

In small and large animal validation studies[3,5] the prototype zwitterionic NIR fluorophore ZW800-1 exhibited renal-exclusive clearance and elimination of the entire injected dose into urine over a period of 4–6 h. While passing from the kidneys to the bladder, ZW800-1 provided exquisite visualization of the ureters, including structure (i.e., anatomical traverse) and function (i.e., flow and patency).

Iatrogenic ureteral injury is amongst the most feared complications of lower abdominal surgery, with incidence varying from 0.5–1% in cancer surgery, to as high as 10% in gynecologic oncologic surgery[6–9]. Ureters are thin-walled and at risk for injury as they are poorly distinguished from surrounding retroperitoneal tissue and are usually in a collapsed state. Unrecognized ureteral damage during surgery can lead to long-term morbidity, including kidney failure. Ureteral stent placement is often used to decrease iatrogenic damage, but is associated with complications and can itself result in iatrogenic ureteral damage[10]. NIR fluorescence imaging could potentially add value during abdominopelvic surgeries by providing non-invasive, real-time visualization of the ureters even before surgical exploration[11,12].

In this study, we present the complete clinical translation of ZW800-1, the first zwitterionic NIR fluorophore, and demonstrate its utility in visualizing and assessing ureter structure and function during laparoscopic lower abdominal surgery within 10 min after IV administration, without altering the look of the surgical field.

## Results

**ZW800-1.** ZW800-1 (molecular formula $C_{51}H_{66}N_4O_9S_2$; molecular weight 943 Da) is a small zwitterionic molecule with peak absorption of 770 nm, an extinction coefficient at peak absorption 253,900 $M^{-1}cm^{-1}$, peak emission of 788 nm, and a quantum yield in serum of 15.0%. It was manufactured as a sterile, lyophilized powder under Good Manufacturing Practices (GMPs) in the GMP Facility of Leiden University Medical Center, The Netherlands.

**Preclinical studies.** In off-target assays, ZW800-1 showed no significant inhibition of 44 selected targets recommended by four major pharmaceutical companies[13]. No evidence of genotoxicity was observed in the in vitro bacterial mutation assay, in vivo micronucleus assay using hematopoietic cells, or Comet assays for the assessment of DNA strand breakage in liver cells. The no observable adverse event level (NOAEL) in rats was 24.5 mg/kg, with the human equivalent dose (HED)[14] set at 3.95 mg/kg. The maximum-tolerated dose in rats was 1000 mg/kg. The cardiovascular and respiratory study in dogs indicated that administration of a single IV 30 min infusion of ZW800-1 at doses of 0.7 and 7.0 mg/kg does not elicit any effects on the cardiovascular, pulmonary, or body temperature parameters.

The preclinical data in rats and pigs[3,5] suggested that ZW800-1 is pharmacologically inert, and the human dose for adequate ureter visualization would be in the range between 0.5 and 5.0 mg. To be conservative, we opted to start with the lowest dose in this range, instead of a higher dose permitted by the toxicology. We then increased the dose to 2.5 mg and then 5.0 mg during the phase I study. These doses were supported by the toxicology findings, which suggested that even a starting dose of 27.65 mg for a 70 kg adult could be justified (10% of the HED derived from the rat NOAEL, which was 3.95 mg/kg)[15].

**Clinical studies: safety and tolerability.** A total of 28 subjects (16 healthy volunteers and 12 patients) were enrolled in the study (Supplementary Table 1). There were no serious adverse events (AEs) attributed to ZW800-1. Those AEs reported during the trial were mild or moderate, none required interruption of the trials, and all resolved without sequelae. Within the healthy volunteer group, seven volunteers (one received placebo) reported a total of ten AEs. In the patient group, three patients experienced a total of four AEs, all unrelated to ZW800-1. The reported AEs were associated with the post-operative course of the undertaken surgery. A detailed listing of reported AEs is provided in Supplementary Table 2. No trends or changes of clinical importance were observed in the vital signs, clinical laboratory tests, or electrocardiograms after dosing in both groups. Overall, all the administered doses (0.5, 1.0, 2.5, and 5.0 mg) were tolerated well. Doses up to 5.0 mg did not elicit any acute toxicity, nor any hypersensitivity reactions.

**Clinical studies: pharmacokinetics.** The pharmacokinetic (PK) results in blood acquired in both studies were consistent (Supplementary Fig. 1). PK analysis showed that ZW800-1 fluorescence was measurable in serum up to 24–48 h post dose. No differences were observed between males and females. Of the 16 included healthy volunteers, two volunteers (one in the 2.5 mg dose level and one in the 5.0 mg dose level) were completely excluded from the PK analysis due to subcutaneous infusion of the study drug. Even though the IV line appeared to be inserted correctly, fluorescence imaging demonstrated a fluorescent spot at the infusion site. Subcutaneous infusion did not lead to local side effects. Blood PK data from one volunteer (2.5 mg dose level) was excluded because blood was erroneously drawn from the ZW800-1 infusion cannula, tampering with the accuracy of the data. All of the PK blood results of the patients were included in the analysis. NIR fluorescence imaging of the skin in healthy volunteers was concordant with the pharmacodynamic measurements (i.e., dose–response relationship) and showed increasing SBR with higher doses (Supplementary Fig. 2). After an initial vascular flush that highlighted arteries, capillaries, and veins approximately 8–10 s after intravenous bolus injection (Supplementary Movie 1), which is similar to that seen using ICG, the maximum SBR in skin was observed approximately 2 h post injection in all

three dosing cohorts, and declined to baseline at 24 h post injection (Supplementary Fig. 2), suggesting extravasation and reabsorption.

In the healthy volunteers, the cumulative urine excretion of ZW800-1, expressed as the percentage of the injected dose, decreased as the dose increased (Supplementary Fig. 3). The lowest dose group (0.5 mg) had an average cumulative excretion of 93% ZW800-1 at 12 h post dosing. In the higher dose groups (2.5 and 5.0 mg), the average cumulative excretion of ZW800-1 at 12 h post dosing were 70 and 63%, respectively. The decrease in excretion in the higher doses might be explained by the breakdown of ZW800-1 over time, which is known to occur by serum proteins in blood[16]. Other clearance routes are unlikely, as the calculated clearance rates correlate with the human glomerular filtration rate. The majority of ZW800-1 was cleared within the first 2 h after administration. Based on the excretion of approximately 40% of the injected dose of ZW800-1 within the first hour after intravenous injection into human volunteers, we hypothesized that optimal visualization of the ureters intraoperatively would occur at a dose of 2.5 mg administered approximately 5–10 min before ureter visualization is desired.

In the patients, the amount of urine collected was noticeably variable and considerably more concentrated than the healthy volunteers, making the two groups (healthy volunteers and patients) incomparable to each other. There were major differences in fluid intake and urine excretion between the healthy volunteers and patients undergoing laparoscopic surgery, explaining the intersubject variability of the PK urine results within the groups. The variability in urine production can be explained by the physiological effects during surgery. It is known that during laparoscopic surgery, the urinary output decreases due to the created pneumoperitoneum. Due to the significantly decreased and inconsistent urine production during surgery, the cumulative urine excretion for the patients could not be measured accurately.

**Assessment of ureter structure and function**. During laparoscopic surgery, ZW800-1 was administered once the surgeon had identified the location of the ureters under white light, either fully exposed or still covered by peritoneum. During and after administration, the ureters were observed with the NIR channel on the imaging system. ZW800-1 produced high-quality images on all surgical imaging systems tested, including the Olympus®, the Da Vinci® robot, and the FLARE® MIS system (Fig. 1). Importantly, the ureters could be identified while residing under peritoneum (Fig. 2) or even under eschar produced by cauterization during cancer resection.

Tissue perfusion was visible directly after administration and the ureters became fluorescent within 10 min (ranging from 2–10 min) after ZW800-1 injection in all patients (1.0, 2.5, and 5.0 mg). With

1.0 mg ZW800-1, the ureters were clearly visible in the first hour after administration, with a mean SBR of 2.3. With 2.5 mg ZW800-1, the visibility of the ureters was comparable to 1.0 mg ZW800-1, with a slightly stronger signal (SBR of 2.7) in the first hour. However, in the second hour the signal of 2.5 mg was considerably higher when compared to 1.0 mg (2.1 vs. 1.4; unequal variance $t$ test: $p = 0.06$). The signal decreased slightly as time passed; however, with 2.5 mg the ureters were still clearly visible after at least 3.6 h, which was the longest surgical procedure included in the study. In both the 1.0 and 2.5 mg dose, background fluorescence was negligible enabling the surgeons to perform parts of the procedure in the NIR channel, in conjunction with the highlighted ureters. In patients who received 5.0 mg ZW800-1, background fluorescence was perceptibly more evident in the surrounding tissue, organs (i.e., colon), and major blood vessels. This hindered the visibility of the surgical field in the NIR channel and resulted in a significant lower mean SBR within the first hour post dosing compared to 2.5 mg (1.6 vs. 2.7; unequal variance $t$ test: $p = 0.003$; Fig. 3) and 1.0 mg (1.6 vs. 2.3; unequal variance $t$ test: $p = 0.01$; Fig. 3).

Distinguishing ureters can occasionally be challenging where surgeons initially mistakenly identify other structures as the ureter (see Supplementary Movie 2). In this movie, a case is shown where the performing surgeon thought he had identified the ureter. However, during inspection, once ZW800-1 was administered, fluorescence revealed that the ureter was located directly under the initial structure. Mistaking the ureter does not necessarily result in iatrogenic injury, as dissection and isolation is typically done, but unnecessary time can be lost in dissecting the wrong structure. ZW800-1 guidance can aid surgeons in locating the correct structure faster and potentially revise an initial approach.

Not only does ZW800-1 provide visualization of the structure but simultaneously allows real-time assessment of ureter function, and if needed, image-guided repair as it also visualizes its motility. Due to ureteral pulsations, the emitted fluorescence permits clear visualization of the motility and patency of the ureter (Fig. 4; see also Supplementary Movie 3). This time-lapse video shows pulsations of urine down the ureter over a 2 s interval.

## Discussion

Technical advances in the field of surgery, such as laparoscopy and surgical robots, have led to shorter hospitalization and faster recovery[17]. Despite successful implementation, it has not reduced the incidence of visceral injuries. With the use of laparoscopic techniques, surgeons have lost the ability to physically palpate tissue or vital structures during surgery. This study demonstrates that zwitterionic NIR fluorophores like ZW800-1 and NIR fluorescence imaging devices can provide more precise

| Olympus® | Da Vinci® Robot | FLARE® |

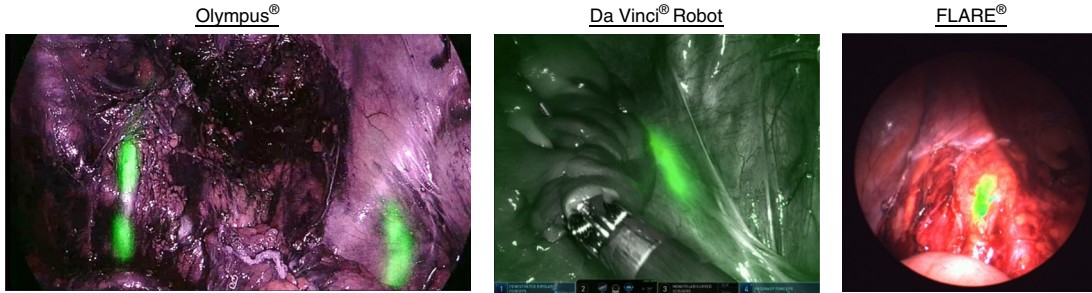

**Fig. 1** High sensitivity detection of ZW800-1 near-infrared (NIR) fluorescence in patients using three different commercial imaging systems. Invisible NIR fluorescence of ZW800-1 is pseudo-colored in green and overlaid in real-time onto the anatomical images

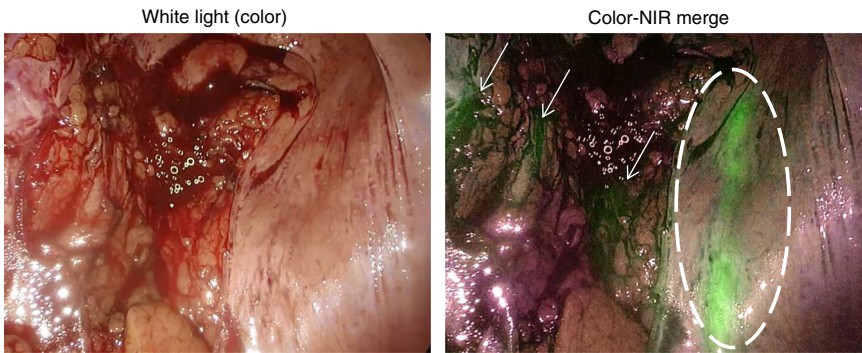

White light (color)                Color-NIR merge

**Fig. 2** Ureter residing under peritoneum. The ureter (dashed circle) is captured during a pulse of urine flow during surgery. The darker green areas in the near-infrared (NIR) fluorescence image (small arrows) is background fluorescence caused by the vessels in the surrounding tissue. The images were acquired using the Olympus® imaging system during laparoscopic surgery

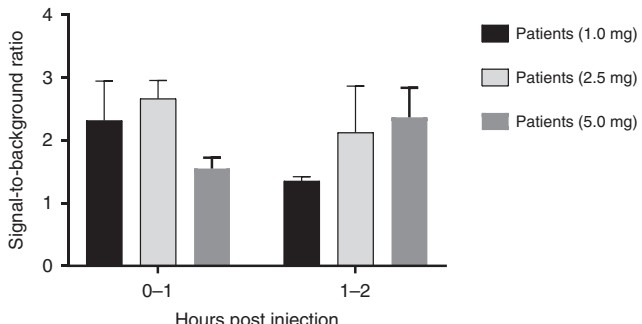

**Fig. 3** Signal-to-background ratio (SBR) of the ureter. Note: the SBR of 5.0 mg is significantly lower than the SBR of 2.5 mg (unequal variance t test: $p = 0.003$) and 1.0 mg (unequal variance t test: $p = 0.01$) in the first hour post dose. The values represent the mean ± SD ($n = 4$ patients each group). Source data are provided as a Source Data file

and safer surgery by visualizing ureter structure and function in real time, even when not yet fully exposed.

Methylene blue (MB) and ICG are currently the only NIR fluorescent agents approved (typically for non-NIR indications) by the Food and Drug Administration (FDA) and the European Medicines Agency. Ureter visualization with NIR fluorescence has previously been studied with MB[18]; however, it is a less favorable fluorophore due to the suboptimal clearance and fluorescent properties. MB is a blue dye that is simultaneously cleared by the liver and kidneys with a peak emission of 700 nm, resulting in higher tissue autofluorescence and reduced penetration capacity[18]. ICG was approved in 1958 as a dark green dye (i.e., a photon absorber) for liver function tests. Yet, it was only in the late 1990s that the fluorescent properties of this heptamethine indocyanine were exploited for imaging. Imaging with ICG is currently being implemented in different surgical indications, such as demarcation of liver metastasis, bile duct imaging, and perfusion assessment[19,20]. As mentioned previously, ICG is not applicable to ureter imaging because of its hepatic-exclusive clearance. However, for other applications, such as biliary tract imaging, it is ideal[21]. Importantly, the safety profile of hepta-methine indocyanines, such as ICG, is remarkable[22]. The new chemical class of zwitterionic NIR fluorophores were built upon the foundation of ICG, but have added valuable features such as low non-specific binding and uptake in tissue, single-step con-jugation to any targeting ligand, and most important for this study, renal-exclusive clearance and elimination from the body into urine.

ZW800-1 appears to solve several problems in laparoscopic surgery using only a single low dose (1.0–2.5 mg). First, it permits non-invasive, precise structural delineation of the ureters in a patient-specific manner, that is, personalized surgery. Second, ureter identification does not require surgical dissection as it can be visualized with overlying tissue. If ZW800-1 is injected 10 min before identification is needed, the ureters are instantly and constantly identified throughout dissection. Third, ZW800-1 provides a large safety window because an extremely low dose is required. A dose of 1 mg is adequate for up to an hour of imaging and may be repeated if needed. A dose of 2.5 mg provides over 3 h of imaging. Both doses require 10 min, at most, to obtain optimal signal strength. Fourth, ZW800-1 works on most available com-mercial imaging systems, even those optimized for ICG, which means widespread availability for patients. Fifth, ZW800-1 per-mits ureter function as well as structure to be quantified. The ability to assess ureter function (i.e., patency and flow) is of huge importance after repair of iatrogenic or traumatic injury and may even broaden its applicability to demonstrate urethral dysfunc-tion during laparoscopy. Along this same line, we know from animal validation studies that micro-nicks of the ureter are quickly identified using NIR fluorescence imaging, which pro-vides intraoperative identification and repair[11,12]. If an injury occurs during surgery, the surgeon should be able to use ZW800-1 to find the site, repair it, and assess ureter function after the repair. Based on the unexpected cases of subcutaneous infusion in healthy volunteers, it appears that tissue damage related to high local concentrations of ZW800-1 is unlikely. Finally, ZW800-1 was engineered for one-step conjugatability using NHS or TFP esters, creating endless possibilities in terms of developing novel targeting ligands.

ZW800-1 can identify ureters on multiple surgical fluorescence imaging systems, which is a key feature of the compound to accelerate clinical adoption. The excitation wavelengths for the Olympus®, Da Vinci® Firefly, and FLARE® imaging systems used in this study were 710–790, 806, and 760 nm, respectively, while the collected emission wavelengths were 810–920, 826–850, and 780–900 nm, respectively. Being engineered for heptamethine indocyanines such as ZW800-1, FLARE® had the highest overlap with peak fluorophore excitation and emission, but qualitatively all three systems performed well. Additionally, the possibility of visualizing ureters under overlying peritoneum or eschar on these systems is of paramount importance during surgery. During oncologic surgery the formation of eschar is inevitable when resecting vascularized tumors, and limiting unnecessary dissec-tion, if possible, is advantageous. This is also why the high penetration depth of NIR light (due to low absorption and

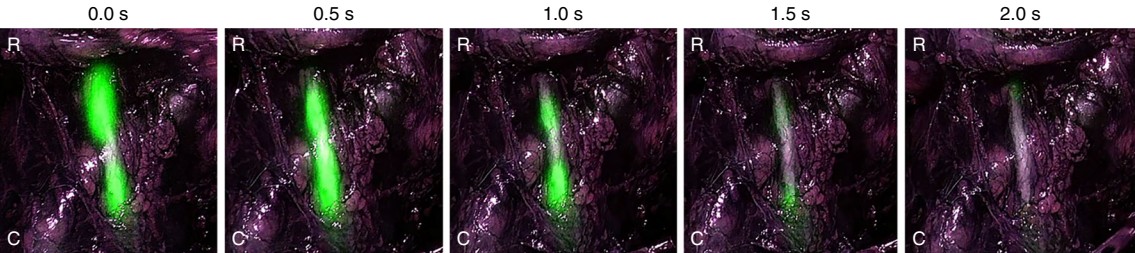

**Fig. 4** Functional assessment of ureter flow and patency with ZW800-1 (acquired using the Olympus® imaging system)

scattering) and ZW800-1's excellent optical properties are important features for image-guided surgery.

The studies presented herein demonstrate the complete clinical translation of a new chemical class from invention, to proof of safety and efficacy, in 8 years. Its package of in vitro, in vivo, healthy volunteer (phase I) and patient (phase II) data can support the design of an appropriately powered phase III pivotal trial. The healthy volunteer study also provided valuable information on safety, PKs, and pharmacodynamics, without exposing vulnerable patients to potentially harmful AEs, and avoiding time-consuming lengthy dose-finding studies in the target population. Interestingly, in the case of nontoxic pharmacologically inert imaging agents such as ZW800-1, the imaging-enabling dose will often be much lower than what NOAEL would suggest[23]. The data obtained in our phase I human volunteer study, for example, resulted in a precise dose estimation for design of an effective phase 2 study using doses far below NOAEL. Furthermore, because of the conjugatability of ZW800-1 and other zwitterionic NIR fluorophores, the development of zwitterionic NIR fluorescence targeted agents should be much more time- and cost-efficient. Currently, the first cancer-targeted molecule using ZW800-1 is being studied in a phase II patient study (European Clinical Trials Database 2017-002772-60).

This translational study suggests that the first-in-class zwitterionic NIR fluorophore ZW800-1 has optimal in vivo and clearance properties, without any observable toxicity. After a single low-dose injection, ZW800-1 provides visualization of the ureters for at least 3 h without altering the look of the surgical field or requiring dissection, and can be used on most commercially available surgical imaging systems. ZW800-1 can also be conjugated to any molecule, including antibodies and other proteins, peptides, and nanoparticles, to create novel targeted NIR fluorescent contrast agents. Taken together, ZW800-1 heralds a paradigm shift in the safety profile of laparoscopic and robotic abdominopelvic surgery.

## Methods

**Preclinical studies.** Preclinical toxicity studies included (1) off-target receptor binding assay (Eurofins Cerep S.A., France); (2) bacterial mutation assay; (3) single infusion range-finding toxicity studies in rats and dogs to support definitive single IV infusion toxicity study in rats (with a genotoxicity and functional observation assessment); (4) single dose toxicity study in dogs; and (5) a single IV infusion study to evaluate cardiovascular and respiratory function in conscious telemetered dogs. The toxicology studies were approved by the National Institutes of Health and conducted by the National Cancer Institute's NExT Program and complied with all relevant ethical regulations.

**GMP production ZW800-1.** The GMP Facility LUMC at Leiden University Medical Center, The Netherlands, manufactured ZW800-1 as a sterile, lyophilized powder. ZW800-1 was prepared in a procedure consisting of seven reaction steps, in which after each reaction the product was purified by precipitation. Each precipitated intermediate was, as an in-process control, checked for identity and purity by ultra-performance liquid chromatography-ultraviolet-mass spectrometry (UPLC-UV-MS). The intermediate drug substance, ZW800-1 before purification, was analyzed with UPLC-UV-MS, nuclear magnetic resonance, thin layer chromatography, and ultraviolet–visible spectroscopy and appearance. Purification of ZW800-1 was performed using preparative gradient reversed-phase high-

performance liquid chromatography (RP-HPLC). Purity of fractions after preparative RP-HPLC purification was controlled using analytical UPLC with UV and MS detection, before pooling the fractions. Absorption spectroscopy analysis on the drug substance pool was performed in order to calculate the volume of ZW800-1 solution to be filled and lyophilized in the final medicinal drug product vials. Identity, purity, content, and general quality control parameters, including appearance, pH, sterility, endotoxins, particles, fluorescence, osmolality, and water, were performed. Stability studies were also performed.

Aseptic pre-administration preparation was done at the hospital pharmacy of Leiden University Medical Center. The vial containing 5.1 mg ZW800-1 was completely dissolved in 5.1 ml glucose 5% infusion fluid by gently swirling the vial by hand, resulting in a clear dark green solution of 1.0 mg/ml ZW800-1 drug product. The reconstituted drug product was stored at 2 to 8 °C until administration. The time between reconstitution and administration varied between 1 and 18 h, as the study drug was prepared by the pharmacy either late in the afternoon the day before dosing or the morning of dosing.

**Phase I: First-in-human study with healthy volunteers.** A total of sixteen healthy volunteers (eight men and eight women) with a median age of 23.5 years (range 18–57 years) participated in this study. The volunteers were considered healthy based on medical screening. A first-in-human, randomized, placebo-controlled study was performed to determine the safety, tolerability, and PKs (in serum, urine, and skin) of ZW800-1. Three dose levels of ZW800-1 (0.5, 2.5, and 5.0 mg) were investigated in three non-overlapping cohorts. Placebo consisted of 0.9% NaCl. The first cohort used a sentinel approach. Subjects were randomized to a 4:2 ratio of ZW800-1 and placebo. In the following two cohorts, the subjects were randomized to a 4:1 ratio of ZW800-1 and placebo. Thus, of the 16 healthy volunteers, 12 received ZW800-1 and 4 received placebo. The study was double-blinded where the investigator, staff, and healthy volunteers were blinded with respect to the treatment until the end of the study. Placebo and ZW800-1 were formulated identically and the syringes were wrapped in aluminum foil. The IV cannulas were covered during dosing and flushed directly after injection with saline. The same independent physician administered ZW800-1 or placebo during the study. Safety and tolerability were assessed by the occurrence of AEs, changes in clinical laboratory parameters, vital signs, electrocardiogram parameters, physical examination, and injection site. At defined time points, blood and urine samples were collected for PK assessment. Moreover, NIR fluorescence imaging of the foot was performed frequently with the Lab-FLARE® Model R1 Open Space Imaging System (Curadel ResVet Imaging, LLC, Marlborough, MA, USA) to assess the perfusion and uptake of ZW800-1 in the skin.

**Phase 2: First-in-patient feasibility study.** An open-label exploratory study was performed in 12 patients (nine men and three women) with a median age of 61 years (range 30–73 years), scheduled for laparoscopic abdominal or pelvic surgery to assess the feasibility and performance of intraoperative NIR fluorescence imaging of the ureters with ZW800-1. All patients received ZW800-1, which was administered via an IV bolus injection during surgery while the patients were under anesthesia. Three doses of ZW800-1 (1.0, 2.5, and 5.0 mg) were evaluated to determine the optimal dose for surgery. Assignment to the dosage groups was based on the order of enrollment in the study (four patients per dose level). The optimal dose was determined by the calculated SBR obtained from the NIR images during surgery; for each hour of surgery, a mean SBR was calculated. Fluorescence imaging was performed with CE-marked NIR fluorescence medical imaging systems such as those made by Olympus® (CLV-S200-IR, The Netherlands) and Intuitive Surgical (da Vinci® Firefly Imaging System, USA), as well as the model FLARE® MIS prototype (Curadel, USA). During surgery the patients were closely monitored for any abnormalities in vital signs or electrocardiogram. At four different time points (0–15, 15–90, 90–300, and 300–720 min), blood samples were obtained and one urine sample at the end of surgery for PK analysis.

Both the healthy volunteer and patient studies were approved by a certified medical ethics review board and conducted in concordance with the Declaration of Helsinki of 1975 (as amended in Tokyo, Venice, Hong Kong, Somerset West, Edinburgh, Washington, and Seoul), ICH-GCP guidelines, and the laws and regulations of the Netherlands. The medical ethics review board Stichting BEBO in Assen, The Netherlands, approved the healthy volunteer study and the medical

ethics review board METC LUMC in Leiden, The Netherlands, approved the patient study. All subjects provided written informed consent prior to the start of any study-related procedure. The healthy volunteer and patient studies are registered in the European Clinical Trials Database under numbers 2016-003919-35 and 2017-001954-32, respectively, as well as in the Netherlands Trial Register under ID NL7209.

**PKs and statistical analysis**. The phase 1 study was designed using commonly accepted subjects per group. For the phase 2 study, the sample size was not based on statistical power considerations due to the exploratory nature of the study. All PK samples acquired in both the healthy volunteer and patient study were measured within 2 h after withdrawal with the Pearl Impulse (LI-COR Biosciences, Lincoln, NE, USA), with an excitation wavelength of 785 nm and emission wavelengths of 805–850 nm. ZW800-1 concentrations in blood and urine were estimated using calibration curves in fresh human blood and phosphate-buffered saline, respectively, and analyzed with PK variable programming dedicated for PK analysis (R 2.12.0 for Windows). The individual ZW800-1 concentration profiles were analyzed using non-compartmental methods. The fluorescence signal in the skin (healthy volunteers) and the ureters (patients) were quantified using the software ImageJ 1.51j8 (National Institute of Health, MD, USA). The SBR was calculated by drawing a region of interest (ROI) around the background area (as baseline) and the fluorescent signal. The ROI around the fluorescent signal was then subtracted from the background. The quantified value of the fluorescence and background were then divided from each other to obtain a SBR. Data are summarized in graphs and bar charts generated by the GraphPad Prism (version 7.0).

**Reporting summary**. Further information on research design is available in the Nature Research Reporting Summary linked to this article.

## Data availability

All study data are presented in the manuscript and supplementary materials. The source data underlying Fig. 3 and Supplementary Figs. 1 and 3 are provided as a Source Data file. Additional raw data that support the findings of this study are available from the corresponding author upon reasonable request.

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

## Acknowledgements

The preclinical work laying the groundwork for this study was supported in part by NIH grant R44-CA-210820, and toxicology was performed by the NCI NeXT program. This study itself was funded by the Dutch Cancer Society (grant UL2010-4732 and UL2012-5561). The content is solely the responsibility of the authors and does not necessarily represent the official views of the National Institutes of Health.

## Author contributions

M.M.D. and B.G.S.M. assisted in both clinical studies. A.R.V., A.G.T.v.S., and J.K. were responsible for the GMP production of ZW800-1. M.J.v.E. assisted in the PK analysis. K.C.P., F.A.H., and R.F.B. performed the surgical procedures. J.V. was the responsible anesthesiologist for the study. J.V.F. developed ZW800-1 and performed the pre-clinical studies. J.B. and A.L.V. supervised the clinical studies. K.S.d.V. and H.J.H executed the clinical studies and wrote the manuscript with assistance from all the contributing authors.

## Additional information

**Competing interests:** J.V.F. is founder and CEO of Curadel, LLC, a for-profit company marketing the FLARE® technology platform for NIR fluorescence-guided surgery. The other authors declare no competing interests.

**Peer Review Information:** *Nature Communications* thanks Eva Sevick-Muraca and the other anonymous reviewer(s) for their contribution to the peer review of this work. Peer reviewer reports are available.

