## [Peer Review File · Nature Communications]

Reviewers' comments:

Reviewer #1 (Remarks to the Author):

The authors describe the use of a zwitterionic near-infrared fluorophores for intraoperative identification of the ureters during laparoscopic pelvic surgery. Zwitterionic NIR fluorophores are novel molecules with geometrically-balanced, electrically-neutral surface charge, which leads to renal-exclusive clearance and ultralow non-specific background binding. In this study, the authors present the first-in-human experience of this new chemical class, as well as the efficacy study in patients undergoing laparoscopic abdominopelvic surgery. ZW800-1 was safe, had pharmacokinetic properties consistent with an ideal blood pool agent, and rapid elimination into urine after a single low-dose intravenous injection. Visualization of structure and function of the ureters started within minutes after ZW800-1 injection and lasted several hours. Zwitterionic near-infrared fluorophores add value during laparoscopic abdominopelvic surgeries and could potentially decrease iatrogenic urethral injury.

The study is novel and the authors are to be commended for their outstanding work and advancement of the field of fluorescence guided surgery. Their findings may set a new standard in patient safety during pelvic surgery.

One minor issue is that one of the authors (JVF) is the CEO of Curadel and lists no competing interests with the studies. Yet it appears from the Curadel website that ZW800-1 is being commercialized by Curadel. Therefore, JVF should disclose any potential competing interests with the studies.

Reviewer #2 (Remarks to the Author):

NCOMMS-18-29928

A novel zwitterionic near-infrared fluorophore ZW800-1 for real time ureter identification

De Valk, et al.

SUMMARY: This communication nicely describes the safety and use of ZW800-1 as a NIR fluorophore with exclusive kidney clearance for ureter identification during laproscopic surgery. Missing key details necessary for an archival journal need to be added and distracting inconsequential passages that are distracting to an otherwise beautiful contribution need to be removed. The following are the major comments that are necessary for publication and should be easily responded to and the more minor comments that the authors should consider to strengthen their contribution.

MAJOR COMMENTS:

1. While the authors provide the peak absorption and emission of ZW800-1, where in the contribution do the authors address the excitation and emission wavelengths probed with each of the various devices (i.e., Flare, Olympus, Da Vinci, Pearl imager). Please add these wavelengths for each of the devices used into the manuscript.
2. Can the authors please describe why the concept of a pharmacological active dose is used? If the safe starting dose is 27.3 mg, why were the actual administered doses 0.5, 1.0, 2.5, and 5 mg?
3. In the second paragraph of the Discussion, ICG is not ideal for ureter imaging (please add) because of its exclusive hepatic clearance, but it may be more ideal for other applications than ZW800-01 due to its rapid clearance from the blood and therefore reduced background.
4. AEs are not reported and should be included in Table 1. The section on “clinical studies: safety and tolerability” appears to be contradictory. In one sentence, it is stated that no subject experienced an adverse event (AE) – yet in the next sentence, 7 out of 16 (that’s almost 50%) reported a total of 10 AEs. These should be reported in the literature, at least in Table 1. The last sentence of the contribution that applauds safety of ZW800-1 isn’t consistent with the adverse events that occurred, but not reported in a Table.
5. The second to last paragraph describing the timing of the experimental plan doesn’t speak to the results presented, is self-congratulatory, and probably should be deleted.
6. The methods describe 16 health controls in a randomized, placebo-controlled study. While the subjects were randomized at a 4:2 ratio, should there not have been 16+8 = 24 subjects if 16 received drug as detailed in the supplemental material?
7. Syringes were wrapped in foil. Is ZW800-1 unstable as ICG is ??? Since the lyophilized dye is reconstituted, what is the timing between reconstitution and administration.
8. The Methods state that NIR fluorescence imaging of the foot was performed with FLARE. But no results are provided? Could the authors actually see the PK blood profile from the foot as is seen with ICG perfusion profiles? The results should report on all methods described and since this imaging was performed, results should be described in the manuscript or supplemental data.
9. In Figure 2 and in the middle of the movie, there is green fluorescence in the middle of the figure and video. In addition in Figure 2 there seems to be a second vessel structure the left of the ureter demarked by the arrow. What is this? This is background? Some comment is needed here. How did the authors exclude these fluorescence regions?

10. Because numerous devices were used, each figure needs to identify the device employed.

MINOR COMMENTS:

1. First sentence of second paragraph on Introduction: please identify what “conventional NIR fluorophores” are.

2. In the section, “Clinical studies: pharmacokinetics” the end of the second paragraph ends with a description of the optical dose and time of injection with the latter not specified in the sentence. Is “time” supposed to relate to the administration during surgery? Time should be specified.

3. Third sentence of the Discussion, shouldn't “NIR fluorescence” be “NIR fluorescence imaging devices?”

4. Many devices were used. IS one better than another?

5. Figure 4 is not very useful. Perhaps a video would better describe what the authors are trying to convey ? Why is there update of ZW8000-1 in the dashed square box?

Reviewer #1

One minor issue is that one of the authors (JVF) is the CEO of Curadel and lists no competing interests with the studies. Yet it appears from the Curadel website that ZW800-1 is being commercialized by Curadel. Therefore, JVF should disclose any potential competing interests with the studies.

We have amended the text to read:

“Competing interests: JVF is founder and CEO of Curadel, LLC, a for-profit company marketing the FLARE[®] technology platform for NIR fluorescence-guided surgery.”

Reviewer #2

MAJOR COMMENTS:

1. While the authors provide the peak absorption and emission of ZW800-1, where in the contribution do the authors address the excitation and emission wavelengths probed with each of the various devices (i.e., Flare, Olympus, Da Vinci, Pearl imager). Please add these wavelengths for each of the devices used into the manuscript.

The excitation and emission wavelengths for all imaging systems used in the study now appear in paragraph 4 of the Discussion as follows:

“The excitation wavelengths for the Olympus[®], Da Vinci[®] Firefly, and FLARE[®] imaging systems used in this study were 710-790 nm, 806 nm, and 760 nm, respectively, while the collected emission wavelengths were 810-920 nm, 826-850 nm, and 780-900 nm, respectively. Being engineered for heptamethine indocyanines such as ZW800-1, FLARE[®] had the highest overlap with peak fluorophore excitation and emission.”

Wavelengths of the Pearl imaging system have been added to the Methods paragraph ‘Pharmacokinetics and statistical analysis’:

“All pharmacokinetic samples acquired in both the healthy volunteer and patient study were measured within 2 hours after withdrawal with the Pearl Impulse (LI-COR Biosciences, Lincoln, NE), with an excitation wavelength of 785 nm and emission wavelengths of 805-850 nm.”

2. Can the authors please describe why the concept of a pharmacological active dose is used? If the safe starting dose is 27.3 mg, why where the actual administered doses 0.5, 1.0, 2.5, and 5 mg?

The following explanation has been added to the second paragraph of Results in the section ‘Preclinical studies’:

“The preclinical data in rats and pigs (3,5) suggested that the pharmacological active dose (15) for adequate ureter visualization would be in the range between 0.5 and 5.0 mg. Administration of these doses would not be precluded by the toxicology findings which suggested that a starting dose of 27.65 mg for a 70 kg adult could be justified as a starting dose in man (10% of the HED derived from the rat NOAEL which was 3.95 mg/kg).(16) We opted for the approach of the pharmacologically active dose and could thus employ conservative study doses of 0.5 mg, 2.5 mg and 5.0 mg for the Phase 1 first-in-human study.”

3. *In the second paragraph of the Discussion, ICG is not ideal for ureter imaging (please add) because of its exclusive hepatic clearance, but it may be more ideal for other applications than ZW800-01 due to its rapid clearance from the blood and therefore reduced background.*

As suggested, the following sentence has been added to the second paragraph of the Discussion:

“As mentioned previously, ICG is not applicable to ureter imaging because of its hepatic-exclusive clearance. However, for other applications, such as biliary tract imaging, it is ideal.(22) Importantly, the safety profile of heptamethine indocyanines, such as ICG, is remarkable.(23)”

4. *AEs are not reported and should be included in Table 1. The section on “clinical studies: safety and tolerability” appears to be contradictory. In one sentence, it is stated that no subject experienced an adverse event (AE) – yet in the next sentence, 7 out of 16 (that’s almost 50%) reported a total of 10 AEs. These should be reported in the literature, at least in Table 1. The last sentence of the contribution that applauds safety of ZW800-1 isn’t consistent with the adverse events that occurred, but not reported in a Table.*

These are excellent points. We have now added **Supplementary Table 2. Overview of adverse events**, which contains a very detailed listing of all adverse events encountered during the Phase 1 and Phase 2 clinical trials. The table is also referenced in the Results section ‘Clinical studies: safety and tolerability’:

“A detailed listing of reported AEs is provided in Supplementary Table 2.”

In addition, we have revised the noted sentence as follows:

“There were no serious adverse events attributed to ZW800-1. Those AEs reported during the trial were mild or moderate, none required interruption of the trial, and all resolved without sequelae.”

5. *The second to last paragraph describing the timing of the experimental plan doesn’t speak to the results presented, is self-congratulatory, and probably should be deleted.*

References to the timing of the experimental plan have been deleted.

6. *The methods describe 16 health controls in a randomized, placebo-controlled study. While the subjects were randomized at a 4:2 ratio, should there not have been $16+8 = 24$ subjects if 16 received drug as detailed in the supplemental material?*

As reported in Methods, the first dose cohort was randomized 4:2 (6 subjects total) and the second and third dose cohorts were randomized 4:1 (5 subjects per dose), thus a total of 16

subjects were enrolled, with 12 receiving ZW800-1 and 4 receiving placebo. We have clarified this point by adding the following sentence:

“Thus, of the 16 healthy volunteers, 12 received ZW800-1 and 4 received placebo.”

This point has also been clarified in Supplementary Table 1.

7. Syringes were wrapped in foil. Is ZW800-1 unstable as ICG is ??? Since the lyophilized dye is reconstituted, what is the timing between reconstitution and administration.

The syringes were wrapped in opaque foil only to permit “double-blinding” of the study. ZW800-1 is green. The placebo is colorless.

To more clearly report the timing of reconstitution and administration, the following has been added to Methods:

“The reconstituted drug product was stored at 2 to 8 °C until administration. The time between reconstitution and administration varied between 1 to 18 hours, as the study drug was prepared by the pharmacy either late in the afternoon the day before dosing or the morning of dosing.”

8. The Methods state that NIR fluorescence imaging of the foot was performed with FLARE. But no results are provided? Could the authors actually see the PK blood profile from the foot as is seen with ICG perfusion profiles? The results should report on all methods described and since this imaging was performed, results should be described in the manuscript or supplemental data.

Skin perfusion imaging results over 24 hours have now been added as **Supplementary Figure 2**. Additionally, the following has been added to Results section titled ‘Clinical studies: pharmacokinetics’:

“NIR fluorescence imaging of the skin in healthy volunteers was concordant with pharmacokinetic measurements, and showed increasing SBR with higher doses. The maximum SBR was observed approximately two hours post-injection in all three dosing cohorts, and declined to baseline at 24 hours post-injection (Supplementary Figure 2).”

9. In Figure 2 and in the middle of the movie, there is green fluorescence in the middle of the figure and video. In addition in Figure 2 there seems to be a second vessel structure the left of the ureter demarked by the arrow. What is this? This is background? Some comment is needed here. How did the authors exclude these fluorescence regions?

Figure 2 has been annotated to denote weakly fluorescent blood vessels near the highly fluorescent, and pulsing, ureter.

10. Because numerous devices were used, each figure needs to identify the device employed.

Each figure now specifies the imaging system used for data acquisition.

MINOR COMMENTS:

1. First sentence of second paragraph on Introduction: please identify what “conventional NIR fluorophores” are.

The first sentence of the second paragraph in the Introduction has been revised to:

“However, a fundamental problem with NIR fluorescence imaging is that conventional NIR fluorophores are polysulphonated, and highly anionic, in order to shield the central hydrophobic resonance structure and improve solubility, and thus exhibit non-specific uptake in tissues and organs after intravenous (IV) injection.”

2. In the section, “Clinical studies: pharmacokinetics” the end of the second paragraph ends with a description of the optical dose and time of injection with the latter not specified in the sentence. Is “time” supposed to relate to the administration during surgery? Time should be specified.

We have clarified this point in the Results section titled ‘Clinical studies: pharmacokinetics’:

“Based on the excretion of approximately 40% of the injected dose of ZW800-1 within the first hour after intravenous injection into human volunteers, we hypothesized that optimal visualization of the ureters intraoperatively would occur at a dose of 2.5 mg administered approximately 5-10 minutes before ureter visualization is desired.”

3. Third sentence of the Discussion, shouldn’t “NIR fluorescence” be “NIR fluorescence imaging devices?”

The third sentence of the Discussion has been revised to:

“This study demonstrates that zwitterionic NIR fluorophores like ZW800-1 and NIR fluorescence imaging devices can provide more precise and safer surgery by visualizing ureter structure and function in real-time, even when not yet fully exposed.”

4. Many devices were used. IS one better than another?

The devices used in the study were qualitatively comparable to each other. To clarify, the following has been added to the Discussion:

“ The excitation wavelengths for the Olympus®, Da Vinci® Firefly, and FLARE® imaging systems used in this study were 710-790 nm, 806 nm, and 760 nm, respectively, while the collected emission wavelengths were 810-920 nm, 826-850 nm, and 780-900 nm, respectively. Being engineered for heptamethine indocyanines such as ZW800-1, FLARE® had the highest overlap with peak fluorophore excitation and emission, but qualitatively all three systems performed well.”

5. Figure 4 is not very useful. Perhaps a video would better describe what the authors are trying to convey ? Why is there update of ZW800-1 in the dashed square box?

As suggested, Figure 4 has been replaced with a video of the case, now labeled **Supplementary Movie 1**. An additional movie showing urine flow in real-time has been added to the manuscript as **Supplementary Movie 2**.

We thank the reviewers for their suggestions, which have strengthened the manuscript considerably. We also thank you for the opportunity to submit a revised manuscript. Please do not hesitate to contact me if you have any further questions or comments.

Sincerely,

Alexander Vahrmeijer, MD, PhD
Oncologic surgeon

Reviewers' comments:

Reviewer #2 (Remarks to the Author):

While the authors have done a good job in addressing the reviewer's comments, two comments still require mandatory revision.

1. The statement that there is a "pharmacological dose" is completely misleading. For example, from the web a definition for "a "pharmacologically active substance" is any active ingredient (its salts, hydrates/crystals and bases) that are easily absorbed by the body via a host of routes of administration AND readily act on various physiological mechanisms w/out any further modification of the medication. I believe that the authors are confusing pharmacological dose with the dose of ZW800-1 that allows imaging. Since pharmacological activity renders a change in a physiological mechanism, and diagnostic contrast imaging agents are supposed to be inert and not pharmacologically active, the authors refusal to clarify what physiologic change occurs due to the administration of a pharmacological dose renders the authors revisions non-responsive to my comment #2. The educated readers of Nature who are familiar with pharmacologically active therapeutics will likewise look to find what the pharmacological activity of ZW800-1 is -- and expect it to be in an archival journal. I suspect that ZW800-1 is pharmacologically inert and that the authors confusion a pharmacological dose with a dose that enables imaging in preclinical studies. A clarification is a mandatory change that needs to be made.

2. In response to my comment, the authors add Supplementary Figure 2 regarding the fluorescence in the skin of the foot following i.v. administration of ZW800-1 and use the images to report signal to baseline (i.e., prior to injection? not described in methods section) ratio (SBR). After reading the methods, I expected the SBR from the skin of the foot to be signal to background (as described in the methods section) and to match the PK profile. Unfortunately, the time-course of the "SBR" for the 5 mg dose peaks at 2 hours in Figure 2 and does not match the PK profile taken from blood. In addition, for ICG, an i.v. administration allows actual imaging of the superficial blood vessels while for ZW800-1, there does not be any demarcation between the vessels and extravascular space from the supplemental figures. Thus, one can conclude that ZW800-1 extravasates. If ZW100-1 extravasates, as it appears to do so from supplemental Figure 2 and the discordant peaks of the SBR and PK profiles, then it is unclear how the authors conclude that the "NIR fluorescence imaging of the skin in healthy volunteers was concordant with pharmacokinetic measurement." The PK profile from blood measurements could be reflecting of change of glomerular filtration rate as well as capillary permeability. While the authors have done a satisfactory job adding the new data, PK modeling could possibly substantiate whether impaired glomerular filtration is responsible for patient PK data. I believe a careful discussion could alleviate the confusion that the educated readers might have.

Response to Reviewer #2

We are pleased to provide the following clarifications and revisions in response to Reviewer #2's comments. Changes to the manuscript are marked in **Red Text**.

Reviewer #2

While the authors have done a good job in addressing the reviewer's comments, two comments still require mandatory revision.

1. The statement that there is a "pharmacological dose" is completely misleading. For example, from the web a definition for "a "pharmacologically active substance" is any active ingredient (its salts, hydrates/crystals and bases) that are easily absorbed by the body via a host of routes of administration AND readily act on various physiological mechanisms w/out any further modification of the medication. I believe that the authors are confusing pharmacological dose with the dose of ZW800-1 that allows imaging. Since pharmacological activity renders a change in a physiological mechanism, and diagnostic contrast imaging agents are supposed to be inert and not pharmacologically active, the authors refusal to clarify what physiologic change occurs due to the administration of a pharmacological dose renders the authors revisions non-responsive to my comment #2. The educated readers of Nature who are familiar with pharmacologically active therapeutics will likewise look to find what the pharmacological activity of ZW800-1 is -- and expect it to be in an archival journal. I suspect that ZW800-1 is pharmacologically inert and that the authors confusion a pharmacological dose with a dose that enables imaging in preclinical studies. A clarification is a mandatory change that needs to be made.

This is an excellent point, so to avoid any confusion by the reader, we have revised the Results section as follows:

“The preclinical data in rats and pigs^{3,5} suggested that ZW800-1 is pharmacologically inert, and the human dose for adequate ureter visualization would be in the range between 0.5 and 5.0 mg. To be conservative, we opted to start with the lowest dose in this range, instead of a higher dose permitted by the toxicology. We then increased the dose to 2.5 mg and then 5.0 mg during the Phase I study. These doses were supported by the toxicology findings, which suggested that even a starting dose of 27.65 mg for a 70 kg adult could be justified (10% of the HED derived from the rat NOAEL which was 3.95 mg/kg).¹⁵”

We have similarly revised the corresponding section in the Discussion:

“Interestingly, in the case of nontoxic, pharmacologically inert imaging agents such as ZW800-1, the “imaging-enabling dose” will often be much lower than what NOAEL

would suggest.²³ The data obtained in our Phase 1 human volunteer study, for example, resulted in a precise dose-estimation for design of an effective Phase 2 study using doses far below NOAEL.”

2. In response to my comment, the authors add Supplementary Figure 2 regarding the fluorescence in the skin of the foot following i.v. administration of ZW800-1 and use the images to report signal to baseline (i.e., prior to injection? not described in methods section) ratio (SBR). After reading the methods, I expected the SBR from the skin of the foot to be signal to background (as described in the methods section) and to match the PK profile. Unfortunately, the time-course of the “SBR” for the 5 mg dose peaks at 2 hours in Figure 2 and does not match the PK profile taken from blood. In addition, for ICG, an i.v. administration allows actual imaging of the superficial blood vessels while for ZW800-1, there does not be any demarcation between the vessels and extravascular space from the supplemental figures. Thus, one can conclude that ZW800-1 extravasates. If ZW800-1 extravasates, as it appears to do so from supplemental Figure 2 and the discordant peaks of the SBR and PK profiles, then it is unclear how the authors conclude that the “NIR fluorescence imaging of the skin in healthy volunteers was concordant with pharmacokinetic measurement.” The PK profile from blood measurements could be reflecting of change of glomerular filtration rate as well as capillary permeability. While the authors have done a satisfactory job adding the new data, PK modelling could possibly substantiate whether impaired glomerular filtration is responsible for patient PK data. I believe a careful discussion could alleviate the confusion that the educated readers might have.

We now understand the confusion of the original text. First, we have changed the term “pharmacokinetics” to “pharmacodynamics” because the point we were trying to make was that NIR fluorescence signal in skin showed the expected dose-response relationship. Second, we have now explicitly stated that ZW800-1 does indeed extravasate, which should avoid any confusion about why peak blood level and peak skin fluorescence occur at different times:

“NIR fluorescence imaging of the skin in healthy volunteers was concordant with the pharmacodynamic measurements (i.e., dose-response relationship) and showed increasing SBR with higher doses (Supplementary Figure 2). After an initial vascular flush that highlighted arteries, capillaries, and veins approximately 8-10 seconds after intravenous bolus injection (Supplementary Movie 1), which is similar to that seen using ICG, the maximum SBR in skin was observed approximately two hours post-injection in all three dosing cohorts, and declined to baseline at 24 hours post-injection (Supplementary Figure 2), suggesting extravasation and reabsorption.”

In the legend of Supplementary Figure 2, we have also clarified what “background” was used for measurement of SBR in skin.

Finally, Reviewer 2 seemed to be hinting that ZW800-1 would be expected to exhibit the same type of fast “vascular flush” as is seen with other NIR fluorophores, such as indocyanine green. We have therefore added Supplementary Movie 1 to anticipate this common question from the reader.

We thank Reviewer #2 for reviewing our data so carefully, and for helping us revise the article to make it more enjoyable for the reader.

REVIEWERS' COMMENTS:

Reviewer #2 (Remarks to the Author):

My concerns have been addressed.